# A Model Based on Clusters of Similar Color and NIR to Estimate Oil Content of Single Olives

**DOI:** 10.3390/foods10030609

**Published:** 2021-03-13

**Authors:** Claudio Fredes, Constantino Valero, Belén Diezma, Marco Mora, José Naranjo-Torres, Manuel Wilson, Gabriel Delgadillo

**Affiliations:** 1Departament of Agricultural Science, Universidad Católica del Maule, Curicó 3480112, Chile; 2Laboratory of Technological Research in Pattern Recognition, Faculty of Engineering Science, Universidad Católica del Maule, Talca 3480112, Chile; m.wilson.hernandez@gmail.com (M.W.); gabrielignaciod@hotmail.com (G.D.); jnaranjo@ucm.cl (J.N.-T.); 3Laboratorio de Propiedades Físicas (LPF_TRAGRALIA), ETSIAAB, Universidad Politécnica de Madrid, 28040 Madrid, Spain; constantino.valero@upm.es (C.V.); belen.diezma@upm.es (B.D.); 4Department of Computer Science and Industries, Faculty of Engineering Science, Universidad Católica del Maule, Talca 3480112, Chile

**Keywords:** infrared spectroscopy, visible image, support vector machine, olive quality

## Abstract

Lipid extraction using the traditional, destructive Soxhlet method is not able to measure oil content (OC) on a single olive. As the color and near infrared spectrum are key parameters to build an oil estimation model (EM), this study grouped olives with similar color and NIR for building EM of oil content obtained by Soxhlet from a cluster of similar olives. The objective was to estimate OC of individual olives, based on clusters of similar color and NIR in two seasons. This study was performed with Arbequina olives in 2016 and 2017. The descriptor of the cluster consisted of the three color channels of c1c2c3 color model plus 11 reflectance points between 1710 and 1735 nm of each olive, normalized with the Z-score index. Clusters of similar color and NIR spectrum were formed with the k-means++ algorithm, leaving a sufficient number of olives to perform the Soxhlet analysis of OC, as reference value of EM. The training of EM was based on Support Vector Machine. The test was performed with Leave One-Out Cross Validation in different training-testing combinations. The best EM predicted the OC with 6 and 13% deviation with respect to the real value when one season was tested with itself and with another season, respectively. The use of clustering in EM is discussed.

## 1. Introduction

The color of each olive has always been valued by farmers in order to estimate the optimal harvest time [1], because of its close relationship with the quantity and quality of the oil. However, fruit skin and flesh color-based maturity indexes (MI) do not always evolve linearly over time, and are affected by local conditions and cultivar characteristics [2]. The classical maturity index (MI) takes into account the sequential colors: green, yellow, red, purple, and the several blacks, from the skin to drupe as a harvest criterion. Color properties of olives during these maturity stages have been characterized using image analysis and a combination of parameters from the Red, Green and Blue (RGB) plus Hue, Saturation and Brightness (HSB) color spaces [3]. In Ref [4], an automated MI using Computer Vision was developed, reaching an R^2^ of 91%. The colors of the olive images have been classified using estimation models (EM) based on support vector machines (SVM) [5]. For enhanced estimation of the ripening index, in other cases machine vision was combined with artificial neural network (ANN) algorithms fed with a set of chemical parameters (oil content, sugar content and phenol content) obtained by historical data for the region where the MI needs to be predicted [6]. Currently, color has not stopped being used to build models for estimating oil content (OC), as the first classification criterion [5,6,7], although the color is not enough to explain the OC as well as the NIR spectrum can explain it [8,9,10].

NIR technology has a high penetration in laboratories to estimate OC in crushed olives, but not in individual olives due to the high variability of the individual fruits [8]. The Partial Least Square regressions (PLS) or principal components regression has allowed managing the variables that affect the OC, such as a number of samples, olives maturity [11], variety [12], fresh weight [8], olive size, and season of harvest [10], among others. Another problem with the EM of OC of individual olives is that the heteroscedasticity assumption is frequently breached, that is, the variance of errors is not regular in all the observations of each olive, ultimately threatening the performance of the EM. To optimize the management of variability in the OC estimation, a progressive clustering by similar color first, NIR spectrum after, and OC finally has significantly improved the prediction error of the EM [8].

The official measurement of the OC using traditional Soxhlet methodology [13] needs more than a single olive to be properly determined, and thus it is necessary to require a batch of them, regardless of the individual characteristics of each olive in the group, because the individuals are ground and mixed [13]. When one wants to know the OC of individual olives, the Nuclear Magnetic Resonance (NMR) and/or the micro-Soxhlet method gives good results [8,10,14], but most oil mills do not have this type of microanalysis, due to its high human and technical resource requirement; furthermore, these are not official methods. Soxhlet method still being the only “ground of truth” for oil mills [15], so that, in the end, traditional Soxhlet method should be used to construct EM of OC, even for individual olives.

In this study, non-destructive mathematical clustering by homogeneity allows maintaining the characteristics of each olive of a batch that after are ground and mixed during the Soxhlet extraction. We proposed that OC of individual olives can be estimated with the traditional Soxhlet method from the cluster of olives, previously measured and grouped by color and NIR homogeneity, as drawn in Figure 1. The objective was to analyze EM of OC of individual olives, based on clusters of similar color and NIR in two seasons.

The structure of this paper is as follows. Section 2 presents the materials and methods used in this research. Section 3 shows the experiments performed and presents the results obtained. Section 4 shows the results analysis discussion. Section 5 presents the patent of the procedure proposed.

## 2. Materials and Methods

This article introduces a novel clustering method to estimate the oil content of individual olives. The innovation consists of forming clusters of similar olives to assign the oil content of the whole cluster to each olive belonging to the cluster. To carry out this, the olives from which the oil is to be extracted must be similar. For this purpose, a process of clustering similar olives was carried out using a NIR descriptor. The clustering procedure is based on the k-means++ algorithm [16], which groups descriptors according to similarity. The advantage of the k-means++ method over the traditional k-means algorithm is an improved accuracy and speed due to a randomized seeding technique for initiating the clusters [17]. The clustering procedure allows to find the maximum number of likely groups, each group having a certain minimum size. The procedure performed in this study is shown in Figure 1 and is as follows:Fresh olives are collected.Olives are grouped by color and NIR similarity using k-means++ clustering.Measurement of the real OC of the cluster and its assignment to each olive in the cluster.Training and testing of EM of OC of similar individual olives that have the same cluster OC as the reference value.

### 2.1. Obtaining the Color and NIR Characteristics of the Olives

Between March and June of the 2016 and 2017 seasons, flawless olives of all colors and sizes were randomly collected from all parts of trees from two rows marked from the super-intensive olive grove (Olivas Don Rafael”, (coordinates −35.1159017, −71.256272), Maule Region, Chile. The samples were picked up once a week for 5 consecutive weeks during the harvest period and transported to the Laboratory for Technological Research in Pattern Recognition of the Universidad Católica del Maule (Chile), where they were arranged in 24-hole trays to measure their color and NIR characteristics on the same day. The day of the harvest, intact olives were randomly located in the holes marked on the trays. After that, olive trays were photographed with a Sony Alpha a58 SLR camera placed in a controlled environment with diffuse halogen lighting inside, to get color images of each olive. Once images of olives were acquired, the NIR spectrum of each identified olive was measured three times at the equatorial section of each olive. An Ocean Optics spectrometer was used in the spectral range of 900–2200 nm with 512 spectral points with an InGaAs array detector in reflectance mode. Color and NIR characteristics of olives were acquired five times in the season; at the end, a number of 25 identified 24-hole trays of measured olives (600 olives/week) were immediately frozen at −20 °C to complete a set of 3000 frozen olives that were waiting to be assigned to a specific cluster each season.

The processing of the images included: the insolation of each olive’s images; the elimination of errors, such as stakes and imperfections attached to the skin of the fruit; and the image binarization to black/white with the OTSU algorithm [16]. Then, using morphological structuring elements with circular closures [18], the images with defects were eliminated (Figure 1). Afterward, the images were converted from RGB to c1c2c3 color model, which is invariant to lighting, to prevent changes in the color value due to possible lighting variations [19]. The mathematical formula of the color model c1c2c3 is presented below:(1)$$c1\;=\frac{R}{\max\left(G,B\right)};\;c2\;=\frac{G}{\max\left(R,B\right)};\;c3\;=\frac{B}{\max\left(R,G\right)}$$

The processing of NIR spectrum consisted of an average of three NIR measured per olive in the equatorial zone. The selection of 11 spectral points every 2.53 nm between 1710 and 1735 nm, based on the report by [11,20], indicated that the spectral absorption of lipids in fresh olives is around 1725 nm, which is the spectral region of interest in this study. Furthermore, to compare the spectra of the 2016 and 2017 seasons, a principal component analysis was performed [21].

### 2.2. Grouping of the Olives for Their Determination of Oil

The clustering of olives was constructed utilizing 14 descriptors corresponding to the c1c2c3 channels of the color model and 11 NIR spectral points between 1710 and 1735 nm, obtained from a data base of NIR and color of each olive. The descriptors were normalized according to the z-score indicator (*Z*), which were calculated as the difference with the mean in respect to the standard deviation of the data sets.
(2)$$Z=\frac{x-\bar{x}}{sD}$$

Clustering was performed by increasing the depth levels, respecting the minimum level of 30 olives per group at each level, which is enough to do a sample and counter sample of OC by the Soxhlet analysis. The clustering was performed by calculating a representative point for each group to be obtained (centroid), based on measures of similarity between these points using k-means ++ algorithms that identify k as the number of centroids and allocates every data point to the nearest cluster, while keeping the centroids as small as possible [17,22]. The information of each olive allowed forming the cluster in a manual way from the frozen trays of olives. The diversity of olives collected allowed for the change from the trays to bags with a minimum of 30 olives for the Soxhlet analysis.

The OC of each cluster of olives was carried out with a six-unit automatic Soxhlet extractor measuring six samples per day. The oil extraction was based on the Soxhlet method of the American Oil Chemists Society [13] and consisted of grinding the fresh olives with a hand homogenizer, weighing, and drying to get between 3 and 5 g of homogeneous dry mill. The samples were put into paper cartridges and then introduced to the Soxhlet extraction units. After 6 hours of extraction, the samples were cooled in a glass desiccator with a porcelain plate and then gravimetrically weighted with a precision balance to obtain the oil percentage based on the dry matter [13]. In the Appendix A section of this article, the worksheet for the oil analysis per cluster of olives in the 2016 and 2017 seasons (S1, S2) can be found. The number of olives per cluster allows measuring OC two times. The OC means of each cluster were assigned as OC reference for each olive that belonged to the cluster, according to our hypothesis of similarity.

### 2.3. OC Estimation Model and Validation

The EM was based on Support Vector Machines (SVM), which is a powerful method for solving problems of non-linear classification. This model is based on supervised training techniques with hyper-parameters for its construction, and that resolves two optimization problems associated with the search of vectors [23]. The first problem is to adjust the method’s specific parameters and the second is to find the hyper-parameters of the SVR. In this work, the hyper-parameters were established using the Simulated Annealing heuristic search algorithm (Simulated Annealing) [24]. To search for these parameters, an error and data set were defined, which was the same that generated a smaller error, corresponding to this training set. To test the model’s accuracy, the root mean square error of cross-validation (RMSECV) was considered and expressed in units of this value [25].

Calibration models were evaluated using the cross-validation test LOOCV [25]. Samples were taken from the 2016 and 2017 seasons, by using 70% of the olives for training (E) and 30% for testing (T) in the same season. Additionally, season combinations were considered in the following set of training (E) and testing (T) experiments: (a) E: 100% Olives 2016 and T: 100% Olives 2017; (b) E: 100% Olives 2017 and T: 100% Olives 2016; (c) E: 50% Olives 2017 + 50% Olives 2016 and T: 50% Olives 2017 + 50% Olives 2016; (d) E: 70% Olives 2017 + 70% Olives 2016 and T: 30% Olives 2017 + 30% Olives 2016; (e) E: 80% Olives 2017 + 80% Olives 2016 and T: 20% Olives 2017 + 20% Olives 2016. Sets c, d, and e were trained and tested 10 times by selecting random samples. In these sets, the standard deviation and the percentage of deviation from the real value were considered.

## 3. Results

### 3.1. Treatment Images and Spectrum of Olives

The removal of distractors, such as peduncles and skin-attached imperfections, was performed without deforming the curvature of the objects [16,17] (Figure 2).

Segmentation of the images from the bottom was performed with the c1c2c3 model in c3 channel, as proposed [5,18]. Figure 3 shows a histogram on the c3 channel, highlighting the classifying power of c3 color channel [19].

Figure 4 shows the raw NIR spectra in the seasons 2016 and 2017 and standard normal variate spectra in the seasons 2016 and 2017. In addition, Figure 5 shows the smoothed second derivative spectra in the seasons 2016 and 2017, marking the spectral range corresponding to the wavelengths between 1710 and 1735 nm, considered for the oil estimation in this study. The higher similitude of the spectrum around this wavelength (Figure 4) and the peaks found there (Figure 5) implicated that the NIR points considered are consistent with previous studies associating these peaks with lipids [11,20].

### 3.2. Cluster of Similar Olives

The result of the clustering was the formation of 31 and 29 groups of 30 olives with similar NIR/color characteristics in the 2016 and 2017 seasons respectively, which were analyzed for their OC. The clustering algorithm used was the K-means++ [17], which is more robust than the traditional K-means algorithm and responds better to the initial position of the centroids, allowing a better choice of initial values K-means clusters, and improves clustering, thus avoiding deficient cluster formation [16,17]. The main difference between the two algorithms, K-means and K-means++, lies around in the centroids selection, which performed the clustering. K-means++ eliminates the dependence of the initialization of the centroid of K-means by establishing a procedure to initialize the centroids of the sets before proceeding to apply k-means. With this procedure, a substantial improvement is achieved in the results, reducing the error compared to the original K-means [17].

The bar graphics of Figure 6 show the results of Soxhlet analysis per cluster in the 2016 and 2017 seasons. It can be observed that the variability of the year 2016 was much greater than 2017, although differences between the seasons were expected [10,26].

### 3.3. Estimation Model of Oil from Single Olives

The model based on Support Vector Machine (SVR) allowed building models for estimating the OC of every single olive, in the different data set evaluated, generating an error that the SVR minimizes in each set. The model of the season 2016 was trained with the olives of the 31 clusters of this season and was tested with the olives of the 29 clusters of 2017. Furthermore, the 2017’s model was trained with all its olives and tested with all olives of the 2016 season. The first row of Table 1 shows the errors of these models. The central rows of Table 1 show the lowest errors of the EM trained and tested with 70% and 30% of the olives in the same season, respectively. Finally, Table 1 shows different combinations of training and testing with the olives belonging to both seasons’ clusters. The root mean square error of cross-validation (RMSECV) of the EM of individual olives was calculated as the mean and standard deviation of 10 random tests.

Table 1 shows that the RMSECV results for the 2016 (70% training and 30% testing) and 2017 (70% training and 30% testing) seasons were 3.1 and 3.5, and the real value deviation was 6% and 7% respectively. Besides, for the sets formed by the two seasons (2016–2017), the set with the best performance was the one formed by 80% training 2016–2017 and 20% tests 2016–2017, which obtained a RMSECV 7.21 and an average deviation of a real value of 13%. These results along with the results of the Figure 6 show great differences between seasons.

## 4. Discussion

The color and NIR spectrum of each olive are characteristics that are highly related to its oil content; however, a single olive is not enough to quantify OC with the official Soxhlet method, thus requiring a group of olives. Previous research [8] pointed out that the current batch-based assessment of the OC (determined by Soxhlet) in mills only reproduces 44% of the underlying heterogeneity, despite being the factory standard, however, the incorporation of individual NIR spectra to the model allowed for the increase to 67% explanation of the OC (%) of olives. Clustering by similarity in the EM of OC of individual olives allowed to control the variability of the sample, resulting in a better final performance [8,11]. In this study, the EM-based clustering achieved good performances in each season by itself, however, it practically doubled the error when seasons are combined because these were different.

Figure 7 shows loadings of the three principal components corresponding to Principal Component Analysis (PCA) of processed spectra by standard normal variate. The highest values of loading 3 are located at 1710–1735 nm region (black rectangle) in the PCA of the spectra of 2016 and 2017 seasons. This difference negatively affected the OC estimation of one season based on the other season. Figure 8 show PCA score plot of the first two PCA categorized by season with a segregation line between seasons.

Concerning the inclusion of the season in the model, some reports indicate three seasons of data are required to create a robust model, but other reports with a wide range of situations have created a robust model within a single season by including samples [26]. These authors indicated that wider conditions and situations improve the application of EM. In this study, the marked differences between two seasons lead to think that at least three seasons should be included in the EM.

Another angle of this study is the tools for capturing NIR and color variables of the model, which could be improved by two aspects: (1) using another instrument such as multi-spectral cameras that would allow to simultaneously measure NIR and color, improving its efficiency and applicability; and (2) using a NIR spectral range lower than 1700 nm, which would allow decreasing the price of the instrument [26].

Regarding the criteria to build the clustering descriptor, this study consisted of 21% of color characteristics and 79% NIR characteristics. The color is related to maturity and this, in turn, defines the oil content [1,3,4]. The NIR is related with OC, but there are several spectrums related to OC. There is a high sensitivity of the EM to NIR spectrum selected [27], highlighted the range between 1153 and 1231 nm for OC [8]. Further research should optimize the range of the NIR spectrum and its weight as descriptor of OC. The maturity entails a color and oil change that depends on the characteristics of the variety, soil, and climate of each year [28], thus it is difficult to control multiple variables in one descriptor. When models are based on a single variable for clustering, the error should be directly proportional to the size of the cluster as well as indirectly proportional to the similitude of their members. The direct measurement of the OC in each fruit with MNR or micro-Soxhlet is an option, but when traditional Soxhlet is used for measuring, it is necessary to cluster the fruits, in fact, many EM could not have been made without previously grouping the olives [7,8,9,10,11].

Further research should look into other EM, such as neural networks or others, to improve the EM performance, and/or look into clustering descriptors more related to a reference value that shall be obtained from the same cluster.

## 5. Patents

Method for estimating the oil of individual olives using non-destructive technologies (WO2019041055A1). WIPO (PCT) in February 2021.

## Figures and Tables

**Figure 1 foods-10-00609-f001:**
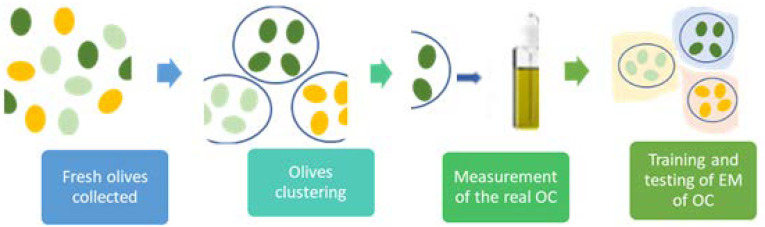
Sequential scheme of the study.

**Figure 2 foods-10-00609-f002:**
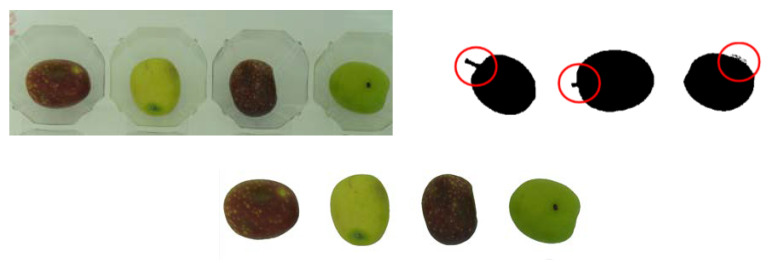
(**Left**) Original images of a group of olives inside a tray; (**Right**) Black/white image with three defective olives. (**Bottom centre**) Final segmentation of group olives.

**Figure 3 foods-10-00609-f003:**
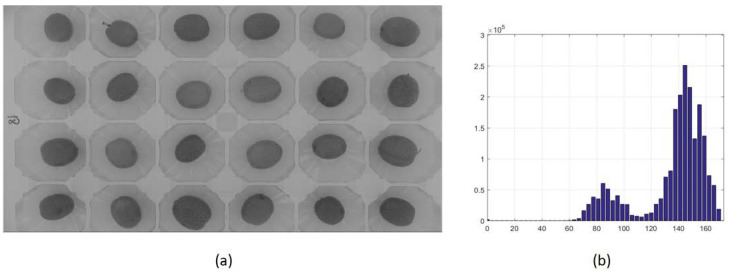
(**a**) Tray of recently collected olives, registered in c3 channel of the c1c2c3 model. (**b**) Histogram of c3 channel for this tray of olives.

**Figure 4 foods-10-00609-f004:**
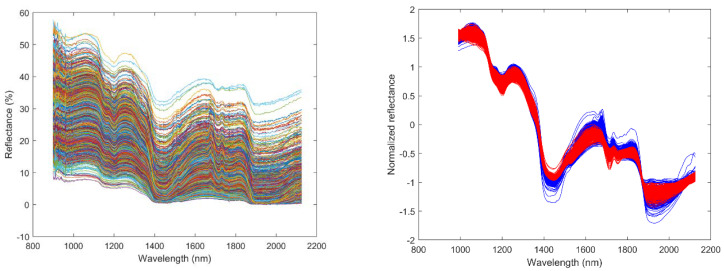
Raw spectra of olives (**left**) and standard normal variate spectra (**right**) in the seasons 2016 (red color) and 2017 (blue color).

**Figure 5 foods-10-00609-f005:**
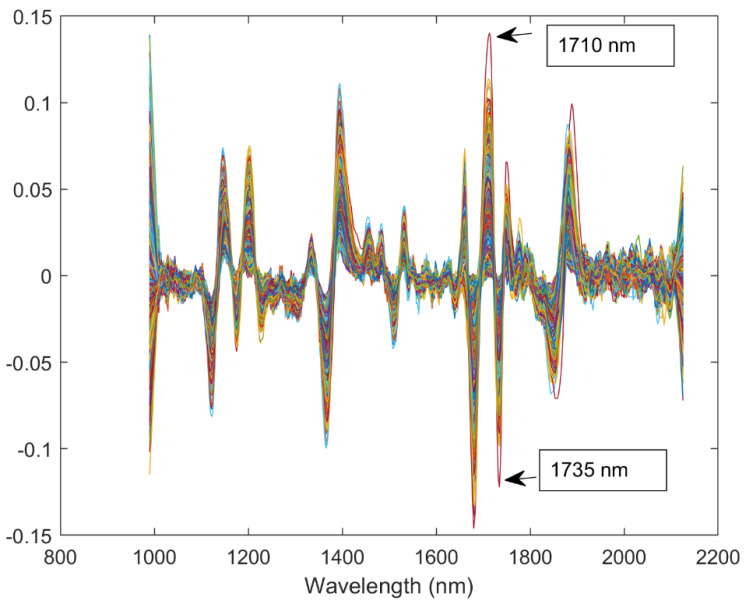
Smoothed second derivative spectra in seasons 2016 and 2017, highlighting the wavelengths used for the clustering.

**Figure 6 foods-10-00609-f006:**
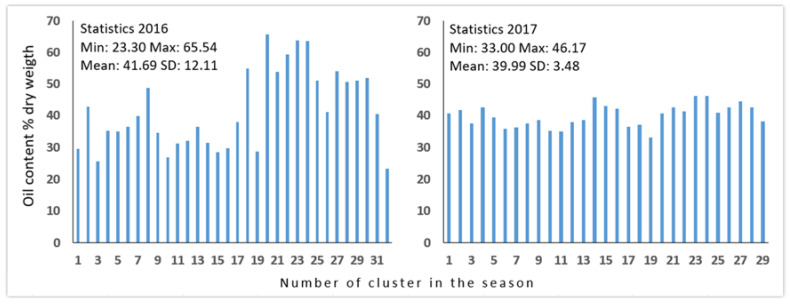
Bar graphics of OC (average of two repetitions) and statistics of the clusters in seasons 2016 and 2017.

**Figure 7 foods-10-00609-f007:**
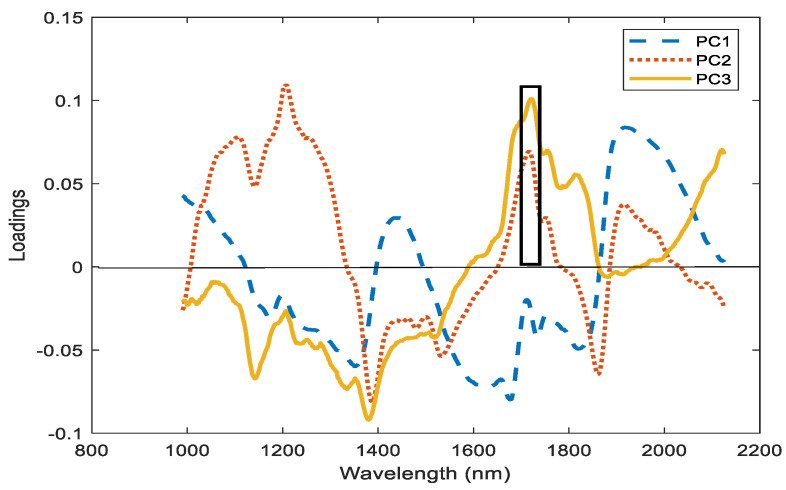
Spectral loadings of Principal Component Analysis of the spectrum of 2016 and 2017 seasons.

**Figure 8 foods-10-00609-f008:**
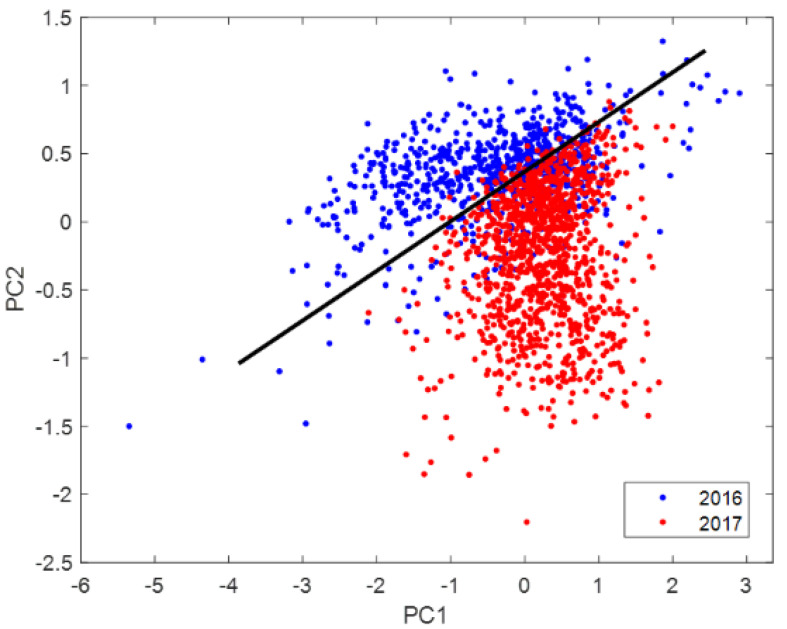
PCA score plot of the first two PCs categorized by season with a segregation line between seasons 2016 and 2017 in both PCA.

**Table 1 foods-10-00609-t001:** RMSECV of EM of OC of individual olives based on clustering by similar color and NIR in different training and testing sets with olives from the 2016 and/or 2017 seasons.

Training Sets 2016/17	Testing Sets 2016/17	RMSECV	
100% Olives 2016	100% Olives 2017	7.35	
100% Olives 2017	100% Olives 2016	8.64	
		**Mean 10 RMSECV**	**SD 10 RMSECV**
70% Olives 2016	30% Olives 2016	3.1	0.4
70% Olives 2017	30% Olives 2017	3.5	0.3
50% Olives 2017 + 50% Olives 2016	50% Olives 2017 + 50% Olives 2016	7.34	0.61
70% Olives 2017 + 70% Olives 2016	30% Olives 2016 + 30% Olives 2017	7.13	0.64
80% Olives 2017 + 80% Olives 2016	20% Olives 2017 + 20% Olives 2016	7.21	0.12

## Data Availability

Data available upon request.

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
