# Peer review of "A Model Based on Clusters of Similar Color and NIR to Estimate Oil Content of Single Olives"

_foods, 2021, doi:10.3390/foods10030609_

Round 1
Reviewer 1 Report
The manuscript is interesting but extensive editing is required, for its language, presentation, description of the method, justification of some of the approaches and better discussion.
Please check the comments of this reviewer added in the pdf file.
Author Response
Dear Reviewer
First and foremost, thank you for taking the time to review my article. I have taken your suggestions and made the appropriate changes according to the comments given.
Lines 72-75 are unnecessary
Thank you for your comment. Dear reviewer, at this point we retain this paragraph in the introduction as a reference guide to the structure of the manuscript. Also, another of the reviewers did not see it initially and requested to add the description of the structure of the manuscript.
Figure 2: the indications of a) and b) must be moved out of the previous image
- I fixed this mistake in the figure 2. Thanks you.
Once again thank you for your time to review our communication.
Reviewer 2 Report
I think this work has a huge interest for the olive oil industry, however i think that more potential can be extracted from it. The experiment desing is appropiate and the instrument is quite useful, however, many things are planned for future works.
Furthermore, I have and one more question..how is possible to do the cluster based on the colour of the olives using an instrument which does not include the visible region?
Line 120: When it comes to NIR measurement, it is interesting to know more information about the measurement process. How many spectra did you acquire? In any specific area of the olive? It could be indicated more information of the instrument used. How many samples do you analyse?
Line 128: If the instrument has a huge spectral range, why is it necessary to remove majority of the range to carry out the treatment? I agree the area that the authors specified are related to lipids, but there are other areas of the spectra with a huge influence in the results, such as around 1400 nm. It would be interesting to see a fast comparative of the results when the whole spectrum of is used.
Line 152: change the expression “optimitization problems”
Line 158: Root mean square error…(RMSECV)..
Figure 4: There are 3 spectra a bit far of the remining. Why is it? I would also like to see the pre-treated spectra
Line 200: which is the difference between K-means and K-means ++?
Figure 5: Introduce label for axis X. Careful! The values of statistics are in Spanish (separation between decimals with ‘,’). Furthermore, in one of them you can see SD and in the other DS.
Table 1: THE RMSECV is in % or the units of the parameter?
I think it is not necessary to try 10 different sets if, after that, the mean value of these 10 is going to be provided. It would be more interesting to make up a calibration and validation sets based on structure method or, in case you want to try 10 different random sets, it would be nice to see the error for each of these and discuss these values.
Why the RMSECV is almost the same when you predict 100% olives 2017 (training set is 100% 2016) that when you include in the training and validation sets different percentages of samples from each year? It sounds strange to me since, in the first case, the totality of the predicted samples is from another season, whereas in the other cases, these samples are supposed to be represented in the training set. It would be interesting to see the distribution of the population in PCA.
Line 226: SD has a real value of 12%.
Line 241-244: This is not totally true. When you work with more than one season, you will have greater representation in your model, and it will improve the prediction of unknown samples in the futures. The problem comes when your seasons are totally different, that’s why it is necessary to visualize them in PCA. Furthermore, there are some works and reviews which talk about this issue:
- Walsh, K., Blasco, J.,Zude-Sasse, M., Sun, X. Visible-NIR ‘point’ spectroscopy in postharvest fruit and vegetable assessment: The science behind three decades of commercial use
Line 246-250: This paragraph is not easy to understand at all. The use of a multispectral camera would enable to measure the colour in the same way it was done in this paper. However, it must be taken into consideration that the acquisition of a NIR imaging system in the range upper 1700 nm would considerably increase the price of the instrument.
Line 266: There is not section number 5.
Author Response
Dear Reviewer,
First and foremost, thank you for taking the time to review my article in deep way. I have taken your interesting and valuable suggestions and we treat to make the appropriate changes according to the comments given.
I have and one more question how is possible to do the cluster based on the colour of the olives using an instrument which does not include the visible region?
We clarified the methodology: The color of the olives was acquired by RGB cameras in a controlled environment (lines 109-118).
I think this work has a huge interest for the olive oil industry, however i think that more potential can be extracted from it. The experiment desing is appropiate and the instrument is quite useful, however, many things are planned for future works.
Furthermore, I have and one more question..how is possible to do the cluster based on the colour of the olives using an instrument which does not include the visible region?
Line 120: When it comes to NIR measurement, it is interesting to know more information about the measurement process. How many spectra did you acquire? In any specific area of the olive? It could be indicated more information of the instrument used. How many samples do you analyse?
Thank you, because your observation aims to order the explanation of the procedure; This information is now explained in the first part of the Methodology that corresponds to the acquiring of COLOR-NIR information from the olives with the VIS cameras and NIR spectrometer with more details (lines 109-118). We wrote first the acquiring of information, and after we wrote the processing of this information (since line 119).
Line 128: If the instrument has a huge spectral range, why is itnecessary to remove majority of the range to carry out thetreatment? I agree the area that the authors specified are related tolipids, but there are other areas of the spectra with a huge influence in the results, such as around 1400 nm. It would be interesting to see a fast comparative of the results when the wholes pectrum of is used.
Thank you for your suggestion. We incorporate several graphs in order to justify our spectral range selected (lines 197-215)
Line 152: change the expression “optimitization problems”
Line 158: Root mean square error…(RMSECV)..(lines 21,244)
We change these words in the text.
Figure 4: There are 3 spectra a bit far of the remining. Why is it? Iwould also like to see the pre-treated spectra
We change the figure 4 and incorporate new figure 5 in order to clarify the spectra.
Line 200: which is the difference between K-means and K-means ++?
This is explained now in lines 223-227 (red words)
Figure 5: Introduce label for axis X. Careful! The values of statistics are in Spanish (separation between decimals with ‘,’). Furthermore, in one of them you can see SD and in the other DS.
Thank you for your comment. The respective changes were made in the statistics and the information of the X axis was added in Figure 5.
Line 246-250: This paragraph is not easy to understand at all. The use of a multispectral camera would enable to measure the colour in the same way it was done in this paper. However, it must be taken into consideration that the acquisition of a NIR imaging system in the range upper 1700 nm would considerably increase the price of the instrument.
Thank you for your comment. The paragraph was modified (red words lines 293-297):
- Table 1 shows that the RMSECV results for the 2016 (100%) and 2017 (100%) seasons were 3.1 and 3.5, and the real value deviation was 6% and 7% respectively. Besides, for the sets formed by the two seasons (2016-2017), the set with the best performance was the one formed by 80% training 2016-2017 and 20% tests 2016-2017, which obtained a RMSECV 7.21 and an average deviation of a real value of 13 %.
We made PCA analysis that show great differences between seasons and it can explain our results (see lines 272-285).
Table 1: THE RMSECV is in % or the units of the parameter?
RMSECV is expressed in units of the parameter, see line 167.
I think it is not necessary to try 10 different sets if, after that, the mean value of these 10 is going to be provided. It would be more interesting to make up a calibration and validation sets based on structure method or, in case you want to try 10 different random sets, it would be nice to see the error for each of these and discuss these values.
Thank you, we made several combinations of training and testing of the model with different set of 2016 - 2017 olives and summarize the results in the Table 1.
Why the RMSECV is almost the same when you predict 100%olives 2017 (training set is 100% 2016) that when you include in the training and validation sets different percentages of samples from each year? It sounds strange to me since, in the first case, the totality of the predicted samples is from another season, whereas in the other cases, these samples are supposed to be represented in the training set. It would be interesting to see the distribution of the population in PCA.
Thank you for your suggestion and I agree with you, we show you and PCA analysis: The Figure 6 shows a Principal Component Analysis of the spectrum of 2016 and 2017 seasons. This difference negatively affected the estimate of one year based on the other year, see the lines 272-285.
Line 226: SD has a real value of 12%. Yes, this value was the SD.
Line 241-244: This is not totally true. When you work with more than one season, you will have greater representation in your model, and it will improve the prediction of unknown samples in the futures. The problem comes when your seasons are totally different, that’s why it is necessary to visualize them in PCA.Furthermore, there are some works and reviews which talk about this issue: - Walsh, K., Blasco, J.,Zude-Sasse, M., Sun, X. Visible-NIR ‘point’spectroscopy in postharvest fruit and vegetable assessment: The science behind three decades of commercial use
Yes, it's true. We add this article to improve the discussion, see lines 287-292.
Line 246-250: This paragraph is not easy to understand at all. The use of a multispectral camera would enable to measure the colour in the same way it was done in this paper. However, it must be taken into consideration that the acquisition of a NIR imaging system in the range upper 1700 nm would considerably increase the price of the instrume.
Thank you, we change the paragraph, see lines 293-297.
Line 266: There is not section number 5.
-Thank you, I fixed it in the paper.
Once again thank you for your time to review our paper.
Reviewer 3 Report
The paper deals with an original topic, the estimatation of the oil content of single olives by color and NIR measurement in combination with statistical methods. But the description of this approach is very difficult to read and to understand especially due to the written English grammar and the sentence structure. For me the paper needs a deep revision to make the challenging and interesting topic readable and understandable for the readers of Foods. The authors should also explain more in detail why such a method is important. The paper has already been published on https://www.preprints.org/manuscript/202012.0405/v1 without reviewing the paper and not comments have been made up to now.
In general:
The authors wrote almost in the whole manuscript from Sohlext, instead of from Soxhlet.
The structure of the manuscript has not to be mentioned in the introduction. It is already given by the journal.
The English language should be deeply revised.
Author Response
Thank you, we made an English revision.
- As can be seen in the image of the https://www.preprints.org/, it is indicated that: This version is not peer-review. Furthermore https://www.preprints.org/ specifies that: “A preprint is a piece of research that has not yet been peer reviewed and published in a journal. In most cases, they can be considered final drafts or working papers”:
- We change the several word Sohlext by Soxhlet in the text. Thank you. It was an involuntary mistake.
- We have modified the text in several paragraphs to explain more in detail why our method is relevant. See red word in lines 18-21 and 82 - 99. Thank you very much for this observation.
- Thank you, but the structure of the manuscript still is at the end of the introduction (see red words 76-78) because another reviewer requests it.
Once again thank you for your time to review our communication.
Round 2
Reviewer 1 Report
The authors have tried improving the article but it is still confusing in the approach and in some choices they made.
Page 5: the error of the reference method should be investigated and reported. There is no indication on whether the results are reported on a “as is” basis, dry matter basis or corrected for a consistent moisture basis of the olives. Was the moisture content assessed in the olives? Please report.
At page 9, the authors have a sentence in Spanish, please remove and check that the references are up-to-date.
At page 10, it is still very difficult to read the table 1, please try to improve it readability and explaining better what is shown.
In the figure at page 10 reporting the histogram of oil content, the standard deviation was not added. Was it because one determination was carried out for each of the samples? Otherwise, please graphically add the standard deviation for each sample.
Discussion: generally, when a model does not perform well on a new season, it means that it has poor robustness. The authors should then build an updated model including the variation coming from the new harvesting year. Why this was not done? I do not see it as useful having a model that only performs well on one season.
It is not surprising that the two seasons show such a large difference when plotting the PCA, but how this issue was solved?
Given the comments in lines 391-395, it seems that the authors are stating that their model is not useful anyway, but I would still be trying building with all possible data and using a validation based on hold-out approach of the data. Please do so and report the differences in terms of results so it can be used as a comparison.
In the discussion section there are some repetitions, please revise.
Also, the authors mention the colour of the olives, but as far as I know there is no relationship between colour and oil content. Is this not true?
Regarding the method used, what was the basis of only using SVM for the classification, and not testing and reporting other methods? I would strongly suggest doing so.
Author Response
Dear Reviewer
Thank you for taking the time to review our article twice.
Your interesting and valuable suggestions have improved your work.
I hope to answer assertively to your suggestions.
"Page 5: the error of the reference method should be
investigated and reported. There is no indication on
whether the results are reported on a “as is” basis, dry
matter basis or corrected for a consistent moisture
basis of the olives. Was the moisture content
assessed in the olives? Please report."
Thank you for the observation. Please read lines 145-155 (blue words without track changes), immediately before 2.3 part: The experiment was based on the Soxhlet method and the results were expressed in dry matter. Furthermore, we added the following text "In the supplementary materials section of this article, it is found the worksheet for the oil analysis per cluster of olives in the 2016 and 2017 seasons (S1, S2)."
"At page 9, the authors have a sentence in Spanish,
please remove and check that the references are up to-
date."
My apologies, I can not find these sentences.
"At page 10, it is still very difficult to read the table 1,
please try to improve it readability and explaining
better what is shown."
Thank you for the observation. Please read lines 239-246 (blue words without track changes), immediately after 3.4 part: we try to explain better what Table 1 shows.
"In the figure at page 10 reporting the histogram of oil
content, the standard deviation was not added. Was it
because one determination was carried out for each of
the samples? Otherwise, please graphically add the
standard deviation for each sample."
Dear Reviewer, thank you. Strictly speaking, the figure does not show histograms, but bar graphs, in which each bar corresponds to the OC average (two Soxhlet analyses) of each group. We change the name histogram by bar graphic. Please read more information in blue words before and after figure 6.
"Discussion: generally, when a model does not perform
well on a new season, it means that it has poor
robustness. The authors should then build an updated
model including the variation coming from the new
harvesting year. Why this was not done? I do not see it
as useful having a model that only performs well on
one season."
It is true that the model loses robustness in two years, but more than that, we attempt to describe an innovative clustering method and discuss how it could work.
"It is not surprising that the two seasons show such a
large difference when plotting the PCA, but how this
issue was solved?
Given the comments in lines 391-395, it seems that
the authors are stating that their model is not useful
anyway, but I would still be trying building with all
possible data and using a validation based on hold-out approach of the data. Please do so and report the differences in terms of results so it can be used as a
comparison."
We know that there is a great difference between seasons, but as the estimation model of the oil, content obtained 6-7% deviation with respect to real value in one season, we believe that is a good approach that deserves to be explained and discussed in this paper.
"In the discussion section there are some repetitions,
please revise."
Thanks you, we made the changes in order to avoid repeating ideas that are already shown. Please see the blue words in the discussion part.
"Also, the authors mention the colour of the olives, but
as far as I know there is no relationship between
colour and oil content. Is this not true?"
Thank you for the observation. Please read lines 299-303 (blue words in the last paragraph without track changes) where we have discussed that theme.
"Regarding the method used, what was the basis of
only using SVM for the classification, and not testing
and reporting other methods? I would strongly suggest
doing so."
Thank you for this important suggestion. Nevertheless, comparing the performance of different models would change the article and it would be necessary to decide between a variety of estimation models based on neural networks and to decide those more appropriate; after, to determine the model parameters and to run them and thereafter to focus on the discussion of model comparison. As we are in the second round of review, we consider that the times are not enough. However, we have placed in the last paragraph of the article (blue words) a reflection on the possibility of using other estimation models.
Thank you again, we have appreciated very much your clever suggestion and we made the changes that we can in the article¡
This manuscript is a resubmission of an earlier submission. The following is a list of the peer review reports and author responses from that submission.